# Towards More Robust NLP System Evaluation: Handling Missing Scores in Benchmarks

## Abstract

The evaluation of natural language processing (NLP) systems is crucial for advancing the field, but current benchmarking approaches often assume that all systems have scores available for all tasks, which is not always practical. In reality, several factors such as the cost of running baseline, private systems, computational limitations, or incomplete data may prevent some systems from being evaluated on entire tasks. This paper formalize an existing problem in NLP research: benchmarking when some systems scores are missing on the task, and proposes a novel approach to address it. Our method utilizes a compatible partial ranking approach to impute missing data, which is then aggregated using the Borda count method. It includes two refinements designed specifically for scenarios where either task-level or instance-level scores are available. We also introduce an extended benchmark, which contains over 131 million scores, an order of magnitude larger than existing benchmarks. We validate our methods and demonstrate their effectiveness in addressing the challenge of missing system evaluation on an entire task. This work highlights the need for more comprehensive benchmarking approaches that can handle real-world scenarios where not all systems are evaluated on the entire task.

## 1 Introduction

Benchmarking and system evaluation are critical processes for assessing the performance of AI systems, providing a standardized means of comparing various models and techniques while keeping track of technological advancements (Ruder, 2021; Dehghani et al., 2021; Post, 2018). However, evaluating general-purpose systems, such as foundation models used for generative tasks (Lehman et al., 2023; Koco et al., 2023b; OpenAI, 2023; Brown et al., 2020; Raffel et al., 2020), presents unique challenges. A single task, metric, or dataset may not be sufficient to effectively gauge their capabilities (Herbrich et al., 2006; Novikova et al., 2018; Sedoc & Ungar, 2020). Therefore, it is crucial to develop tools that can benchmark these systems on a multitude of tasks (Aribandi et al., 2021), enabling a comprehensive assessment of their overall performance (Peyrard et al., 2021).

In recent years, the field of natural language processing (NLP) has made significant strides, with frequent emergence of new models (Lehman et al., 2023; Koco et al., 2023a; Brown et al., 2020; OpenAI, 2023; Raffel et al., 2020; Liu et al., 2019; Fan et al., 2021) and techniques (Bommasani et al., 2021; Hupkes et al., 2022). To evaluate the performance of these systems across various tasks, datasets, and metrics (Colombo et al., 2022c) have been created. However, with the increasing complexity of these benchmarks, missing scores has become a significant challenge. Missing data can arise from a variety of sources, such as benchmarks that are too large or time-consuming to run (*e.g.*, BigBench has recently introduced MiniBench for these reasons (Srivastava et al., 2022)), high costs associated with reproducing experiments (*e.g.*, see Table 3 in Artetxe et al. (2022)), incomplete datasets (see Table 5 in Reid & Artetxe (2022)), data collection errors, data cleaning procedures, data privacy concerns (particularly in-house datasets (Guibon et al., 2021)), and specialized expertise required to process niche datasets (Peng et al., 2019). In recent work, two main approaches have been followed to deal with missing scores, which are discarding data (Pfeiffer et al., 2022) or ignoring certain tasks (see Table 10 in Lin et al. (2022) and Table 5 in Martin et al. (2020)) or evaluations. However, these approaches are unsatisfactory as they can lead to biased and unreliable evaluations.

In this work, we aim to address the challenge of benchmarking NLP systems *when one or several systems cannot be evaluated on a specific task*. We propose the development of effective methods for aggregating metrics that can handle missing data and enable a comprehensive assessment of system performance. Our approach will ensure the reliability and validity of NLP system evaluations and contribute to the creation of benchmarks that can be used to compare and evaluate NLP systems effectively. Specifically, our contributions are listed below.

1. **Introducing a new problem with a direct impact on NLP research:** benchmarking when there are missing system evaluations for an entire task, which has practical implications (Pfeiffer et al., 2022; Lin et al., 2022; Martin et al., 2020; Guibon et al., 2021; Peng et al., 2019).

2. **A novel method for benchmarking NLP systems with missing system scores.** We present a novel method that effectively tackles the issue of missing system evaluations for entire tasks. Our work includes a novel combinatorial approach for imputing missing data in partial rankings. It allows using standard rank aggregation algorithms such as Borda and offers two refinements tailored to the availability of either task-level or instance-level scores of the systems across different tasks.

3. **An extended benchmark for a comprehensive and accurate evaluation of NLP systems:** previous works on score aggregation relied on a benchmark of 250K scores (Colombo et al., 2022b; Peyrard et al., 2021), and did not release the system's input, output, and ground truth texts. In our work, we collected their scores and extended the benchmark by adding over 131M scores.

4. **Extensive validation of benchmarking methods:** Results show that our method effectively handles missing scores and is more robust than existing methods, affecting final conclusions.

## 2 PROBLEM FORMULATION AND STATE OF THE ART

### 2.1 GENERAL CONSIDERATIONS

**Comparing systems with benchmarks.** Benchmarking aims to determine the ranking of systems based on their scores to identify the best-performing systems. In this process, each system is evaluated on individual tests within a larger set and assigned a score according to a specific metric. Depending on the available information, two approaches are typically employed. When only **task-level** information is available (i.e., the system scores on each task), a **task-level aggregation** is utilized to obtain the final ranking. On the other hand, when **instance-level information** is available, i.e., the system scores on each instance of each task test set, an **instance-level aggregation** method is used to obtain the final system ranking. The mean aggregation has been adopted to consolidate information at both the instance and task levels.

**Benchmarking in the presence of missing data.** As benchmarks and models continue to grow in size and complexity, the occurrence of missing system performance of entire tasks becomes increasingly common. This is particularly true in situations where one or more systems cannot be evaluated on a specific task due to factors such as the high cost of running the model or the extensive computational requirements of the benchmarks (Gehrmann et al., 2022a; 2021; 2022b). Fig. 1 illustrates the general framework (*i.e.*, with instance and system level).

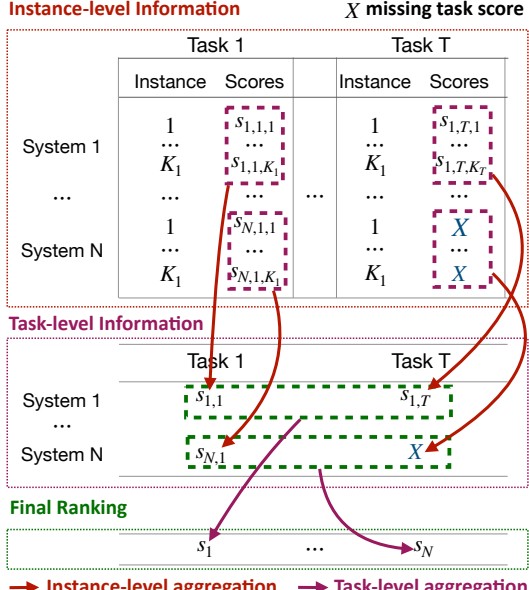

Figure 1: **Framework for benchmarking NLP systems with two information granularity**: *instance-level* (red above) and *task-level* (purple below). The final goal of benchmarking is to produce a ranking (green bottom). The instance-level aggregation allows for the derivation of task-level information, which is used to synthesize system performance via the final ranking (in green). **X** indicates the presence of missing values.

## 2.2 PROBLEM FORMULATION

This discussion will use notation similar to that in the previously mentioned work (Colombo et al., 2022b). In essence, we are dealing with a scenario where $N$ systems are being compared based on their performance on $T$ different tasks. Each task $t \in \{1, \ldots, T\}$ has a specific metric $m_t$ associated with it and has been evaluated on $k$ test instances with $k \in \{1, \ldots, K_t\}$, where $K_t$ is the test size of task $t$. The score of each system on each instance of each test set is represented by the real number $s_{n,t,k} \in \mathbb{R}$. The final goal of benchmarking is to output a ranking of each system according to some objective criterion. We denote by $\mathfrak{S}_N$ the symmetric group on $N$ elements. With this objective in mind aggregating instance and task level information is equivalent to computing a permutation $\sigma \in \mathfrak{S}_N$ corresponding to the final ranking of the $N$ systems. In this formalism, system $i$ is the $\sigma_i$-th best system according to the considered aggregation. Equivalently, ordering $\pi = (\pi_1 \succ \pi_2 \succ \ldots \succ \pi_N)$ denotes that $\pi_i$ is better than system $\pi_{i+1}$ for all $i$. Let us first define the different granularity of benchmarking depending on whether we have access to instance scores.

**Aggregating with Missing Task Level Information.** *Given a set of scores $(s_{n,t}, 1 \leq n \leq N_t, 1 \leq t \leq T)$ where $N_t$ is the number of systems for which we have access to the score on task $t$, find a proper aggregation procedure.*

Thus the problem of task-level information aggregation boils down to finding $f^T$:

$$f^T : \underbrace{\mathcal{S}_{N_1} \times \cdots \times \mathcal{S}_{N_T}}_{T \, times} \longrightarrow \mathfrak{S}_N. \tag{1}$$

where $\mathcal{S}_{N_t} = (s_{n,t}, 1 \leq n \leq N_t)$ is the set of scores achieved by each system evaluated on the task $t$. Note that only $N_t$ systems are evaluated on task $t$.

In many cases, we not only have access to task-level performance but also individual instance-level scores. As a result, the challenge lies in effectively aggregating information at the instance level.

**Aggregating Missing Instance Level Information.** *Given a set of scores $(s_{n,t,k}, 1 \leq n \leq N_t, 1 \leq t \leq T, 1 \leq k \leq K_t)$ where similarly as previously $N_t$ is the number of systems for which we have access to the score on task $t$, find a proper aggregation procedure.*

Thus the problem of instance-level information aggregation boils down to finding $f^I$:

$$f^I : \underbrace{\mathcal{S}_{N_1}^1 \times \cdots \times \mathcal{S}_{N_1}^{K_1} \times \cdots \times \mathcal{S}_{N_t}^k \times \cdots \times \mathcal{S}_{N_T}^1 \times \cdots \mathcal{S}_{N_T}^{K_T}}_{T \sum_t K_t \, times} \longrightarrow \mathfrak{S}_N. \tag{2}$$

where $\mathcal{S}_{N_i}^k = (s_{n,t,k}, 1 \leq n \leq N_i)$ is the set of the score achieved by each system evaluated on the task $t$ for the specific instance $k$.

**Remark 1.** *In the context of complete ranking, which is also a classical setting for benchmarking NLP systems and has been addressed in Colombo et al. (2022b), we have $N_t = N$ for all $t \in [1, T]$.*

## 2.3 HANDLING COMPLETE SCORES IN NLP SYSTEM EVALUATION

The literature relies on two main techniques for aggregating score information to benchmark machine learning systems: mean aggregation and ranking-based aggregation.

**Mean aggregation ($\sigma^\mu$)** is the default choice for practitioners. At the task level $\sigma^\mu$ is defined as $\sigma^\mu = argsort \left( argsort \left[ \frac{1}{T} \sum_{1 \leq t \leq T} s_{n,t} \text{ for } 1 \leq n \leq N \right] \right)$ and at the instance level $\sigma^\mu = argsort \left( argsort \left[ \frac{1}{T} \sum_{1 \leq t \leq T} \frac{1}{K_t} \sum_{1 \leq t \leq K_t} s_{n,t,k} \text{ for } 1 \leq n \leq N \right] \right)$, where $argsort(\mathbf{u})$ is the permutation that sorts the items in $\mathbf{u}$. However, this approach has its limitations, particularly when evaluating tasks of different natures or using evaluation scores that are not on the same scale. Indeed in NLP, metrics can have different ranges (or even be unbounded) and systems are evaluated based on diverse criteria such as quality, speed, or number of parameters. In such cases, conventional rescaling or normalization techniques may not sufficiently capture the inherent difficulty of each task.

**Ranking Based Aggregation** To address the challenges mentioned, researchers have proposed ranking-based aggregations (Peyrard et al., 2021; Colombo et al., 2022b). These methods aggregate rankings instead of scores. In Colombo et al. (2022b), the authors tackle the problem of generating a ranking by aggregating rankings, utilizing the Borda count method (see Ssec. E.1 for more details on Borda Count) known for its computational properties (Bartholdi et al., 1989; Dwork et al., 2001; Ali & Meilă, 2012). Extending the Borda count method is not a straightforward task either. In the next section, we present our aggregation procedure that can handle missing system scores on a whole task.

## 3 RANKING WITH MISSING SYSTEM EVALUATION

In this section, we will outline our methodology for ranking multiple systems in multi-task benchmarks, even if some systems have not been evaluated on one or more tasks. We use the ranking and ordering notation interchangeably.

### 3.1 PARTIAL RANKINGS

**Mapping Scores to Partial Rankings** To address the challenge of benchmarking with missing system evaluations, we propose a ranking-based approach that focuses on aggregating rankings rather than directly combining scores. Suppose we have a specific task $t$ with a task-level score denoted as $S_{N_t}$, or in the case of instance-level information, a task $t$ and instance $k$ with score $S_{N_t}^k$. In scenarios where there are missing evaluations at the task-level or instance-level, a *partial ranking* of systems is generated. A partial ordering represents an incomplete ranking that includes only a subset of items from a larger set. We denote the partial ordering of systems as $\pi^{N_t} = (\pi_1 \succ \pi_2 \succ \ldots \succ \pi_{N_t})$ for the task-level scenario, and as $\pi^{N_t,k} = (\pi_1^k \succ \pi_2^k \succ \ldots \succ \pi_{N_t}^k)$ for the instance-level scenario. Here, $\pi_i$ represents the $i$-th best system according to the set $S_{N_t}$ in the task-level scenario, while $\pi_i^k$ represents the $i$-th best system according to $\pi^k$ in the instance-level scenario.

**Compatible Permutation** When working with partial rankings, it is necessary to construct a complete ranking that respects the order of the evaluated systems, i.e., a linear extension of the partial ranking. This is accomplished by creating a compatible permutation (Gessel & Zhuang, 2018), which is a permutation of all systems consistent with the partial ranking. To construct a compatible permutation, we begin with the partial ranking of the evaluated systems and extend it to include the missing systems while maintaining the order of the evaluated systems. For example, let's consider a partial ordering $\pi_1 \succ \pi_2$ based on the evaluation of only these two systems. If there is an additional system that has not been evaluated, we can construct three compatible permutations: $\pi_3 \succ \pi_1 \succ \pi_2$, $\pi_1 \succ \pi_3 \succ \pi_2$ and $\pi_1 \succ \pi_2 \succ \pi_3$. These permutations ensure that the ordering of the evaluated systems is preserved while incorporating the missing system into the complete ranking.

**Why use a combinatorial approach?** imputing missing data using compatible permutations enables us to leverage the Borda aggregation, inheriting its theoretical and practical advantages. Unlike classical methods like harmonic Fourier analysis (Kondor & Barbosa, 2010; Kondor & Dempsey, 2012; Clémençon et al., 2011) or multi-resolution analysis (Sibony et al., 2015), our approach works, providing a distinct combinatorial solution for imputing missing data in partial rankings.

### 3.2 OUR RANKING PROCEDURES: FROM SCORES TO SYSTEM RANKING

In summary, our method can be described in two steps:

> **Our ranking procedure in a nutshell**
>
> 1. **Matrix Representation of the rankings (Sssec. 3.2.1).** To harness the full potential of the available information in partial rankings, we **efficiently** generate all compatible permutations from the given partial rankings.
>
> 2. **Final System Ranking from Matrix Representation**. To obtain the final ranking of the systems, we propose a one-level ($\sigma^l$) approach (see Sssec. 3.2.2) for both task-level and instance-level information and a two-level aggregation approach ($\sigma^{2l}$) for instance-level information (see Sssec. 3.2.3).

### 3.2.1 MATRIX REPRESENTATION OF THE RANKINGS

**Intuition**. The first step in our algorithm is to summarize the available information in all tasks and to impute the missing information in a consistent manner. To do this, we use a matrix representation $M^\pi$ for each partial ranking $\pi$. This matrix decomposes the ranking information in pairwise variables, i.e., for every pair of systems $i, j$ there is a variable representing the probability that system $i$ outperforms system $j$.

**Why using matrix representation?** Using pairwise information has many advantages in ranking problems with missing data since it allows decomposing the total ranking information in $N(N-1)/2$ different variables. This decomposition has been used in statistical problems on partial and complete rankings (Fürnkranz & Hüllermeier, 2003; Lu & Boutilier, 2014a;b; Shah et al., 2017), for computing distances among partial rankings (Fagin et al., 2003), clustering (Ailon, 2010) and classification (Hüllermeier et al., 2008) among others. However, these problems consider specific forms of missing data such as top-$k$ rankings (Fagin et al., 2003) or bucket orderings (Achab et al., 2019). Our approach differs from the aforementioned literature in the fact that we impute the missing data in a consistent manner in order to be able to deal with arbitrary missing data.

**Efficiently building** $M^\pi$. Let us consider a partial ranking $\pi$ and let $M^\pi \in [0, 1]^{N \times N}$ be its matrix representation. Matrix $M^\pi_{ij}$ denotes the proportion of complete rankings that are compatible with $\pi$ and satisfy the condition $i \succ j$, where $i$ and $j$ are distinct systems in the task. Formally, we can distinguish three cases:

1. *if system $i$ is directly compared to system $j$ in $\pi$*. In this case, we set $M^\pi_{i,j} = 0$ *if* $i \succ j$ *else* $M^\pi_{i,j} = 1$

2. *if no information is provided for either system $i$ or system $j$ in $\pi$*, meaning that both systems are unobserved in the partial ranking. In this case, $M^\pi_{i,j} = 0.5$, which is the natural choice when no information is available.

3. *if we lack direct information about the comparison between system $i$ and $j$ in $\pi$ (one system was evaluated and the was not)*, we represent this situation by setting the corresponding matrix entry to the proportion of compatible permutations ranking system $i$ higher than system $j$ among the total number of compatible permutations (see Ap. E).

A naive algorithm for generating the matrix $M^\pi$ from $\pi$ would have factorial complexity and it is thus exorbitant in practice for a relatively small number of systems, say $N > 10$. **One of the contributions of our solution is to reduce the complexity to $O(n^3)$ by efficiently computing $M^\pi_{i,j}$.** The closed-form expressions for $M^\pi_{i,j}$ as well as the proof for uniformity can be found in Ap. E.

### 3.2.2 SYSTEM RANKING FROM MATRIX REPRESENTATION: A ONE LEVEL APPROACH ($\sigma^l$)

**Intuition.** At this stage, we have demonstrated the construction of a matrix $M^\pi$ for a given partial ranking. However, in benchmarking scenarios, systems are typically evaluated on multiple tasks (in the case of task-level evaluation) or on multiple instances and tasks (in the case of instance-level evaluation). Consequently, it becomes necessary to combine multiple partial rankings. In this section, we will describe our approach for performing the one-level aggregation to address this requirement.

**Combining Multiple Partial Rankings for Benchmarking.** To combine the different matrices into a single matrix $M$ we sum over all the tasks (in the case of task-level information) or instances and tasks (in the case of instance-level information). Formally, this is achieved by performing the following operation to obtain the combined matrix $M^I = \sum_{t \in [1,T]} \sum_{k \in [1,K_t]} M^{\pi^{r_t,k}}$, where $M^{\pi^{r_t,k}}$ represents the partial ranking induced on task $t$ and instance $k$. Similarly, for the task level we define $M^T = \sum_{t \in [1,T]} M^{\pi^{r_t}}$ where $M^{\pi^{r_t}}$ represents the partial ranking induced on task $t$.

**Obtaining the final system ranking** In the final step, our goal is to obtain the final system ranking $\sigma^l$ based on the matrix $M^I$ or $M^T$. To achieve this, we use the Borda Count method, which involves computing the column-wise sum of the matrix and return the permutation that sorts the scores in

increasing order. This step aligns with the approach proposed in Colombo et al. (2022b). Formally:

$$\sigma^l = argsort\left(argsort\left[\sum_i M_{i,0}, \cdots, \sum_i M_{i,N}\right]\right). \tag{3}$$

Here, $M$ represents the matrix $M^T$ for task-level information, and $M^I$ for instance-level information.

### 3.2.3 SYSTEM RANKING FROM MATRIX REPRESENTATION: A TWO-LEVEL APPROACH ($\sigma^{2l}$)

**Intuition.** In the case of instance-level information, we also present a two-step procedure that draws inspiration from the widely adopted two-step mean aggregation approach.

**Procedure.** In the first step, we apply the task-level aggregation approach to generate individual rankings for each task $t$, resulting in $T$ different permutations. In the second step, we aggregate these multiple rankings using the Borda aggregation method. Formally $\sigma^{2l}$ can be computed as:

1. For each task $t$, compute $M^t = \sum\limits_{k \in [1, K_t]} M^{\pi^{r_t,k}}$

2. For each task $t$, compute $\sigma^{2l,t} = argsort\left(argsort\left[\sum_i M_{i,0}^t, \cdots, \sum_i M_{i,N}^t\right]\right).$

3. Compute the Borda count aggregation $\sigma^{2l}$ of $[\sigma^{2l,1}, \cdots, \sigma^{2l,t}, \cdots, \sigma^{2l,T}]$.(Colombo et al., 2022a)

### 3.3 CONFIDENCE INTERVALS FOR $\sigma^l$

When evaluating systems with missing data, it is crucial to measure the uncertainty of partial rankings. In the previous section, we discussed combining partial rankings into a complete ranking. In this section, we analyze the confidence of our data regarding pairwise comparisons of system performance.

Under any ranking model such as Mallows Model (Fligner & Verducci, 1986) or Plackett-Luce (Plackett, 1975), $M_{ij}^\pi$ are random variables of known expected value. What we compute in the previous section is the empirical value of it, $\widehat{M}_{ij}^\pi$ that approximates the true value $M_{ij}^\pi$. Here, we want to know how close these two quantities are. Formally, we are looking for a confidence interval of level $\delta$, that is the value for $c_{ij}$ around $\widehat{M}_{ij}^\pi$ that contains $M_{ij}^\pi$ with high probability, $P(|\widehat{M}_{ij}^\pi - M_{ij}^\pi| \geq c_{ij}) \leq 1 - \delta$. Noting that $0 \leq M_{ij}^\pi \leq 1$, we can use the Hoeffding inequality (Hoeffding, 1994) to compute the value of the confidence interval:

$$c_{ij} = \sqrt{\frac{-\log \delta}{2z_{ij}}}, \tag{4}$$

where $z_{ij}$ is the number of times the systems have been compared.

**Intuition:** to determine the significance of the difference in performance between system $i$ and $j$, we can compare $M_{ij}$ to 0.5. Thus, $i$ performs better than $j$ iff $M_{ij}^\pi > .5$. If the difference between $M_{ij}$ and 0.5 is small, the performance difference between the two systems may not be statistically significant, indicating that we cannot determine which system performs better than the other.

The confidence interval developed above says that the true parameter $M_{ij}^\pi$ is included in the interval $[\widehat{M}_{ij}^\pi - c_{ij}, \widehat{M}_{ij}^\pi + c_{ij}]$ with high probability. It follows that if 0.5 is not in this interval then we can say that one of the systems is better than the other with a high probability. Similar approaches have been proposed to find complete rankings and best-ranked systems with high probability (Busa-Fekete et al., 2014; Szörényi et al., 2015).

### 3.4 BASELINE METHODS

To date, there is no established method for benchmarking NLP systems in the presence of missing data. To compare our proposed algorithm to existing methods, we consider a baseline approach that ignores missing data and relies on mean aggregation. This approach has been used in previous studies (Pfeiffer et al., 2022; Lin et al., 2022; Martin et al., 2020; Guibon et al., 2021; Peng et al., 2019), and we will refer to it as $\sigma^\mu$ in our experiments.

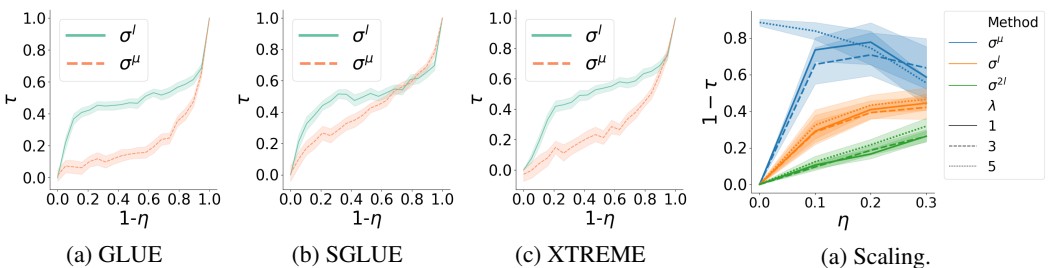

| (a) GLUE | (b) SGLUE | (c) XTREME | (a) Scaling. |

Figure 2: Task-Level Robustness Experiment. We compare the robustness of our method $\sigma^l$ with the mean aggregation method $\sigma^\mu$ by measuring the Kendall $\tau$ correlation coefficient between their respective rankings after removing a proportion $\eta$ of scores and by considering the whole scores.

Figure 3: Synthetic experiment. Robustness for missing data ($\eta$) and different scaling corruptions ($\lambda$).

## 4 SYNTHETIC EXPERIMENTS

### 4.1 DATA GENERATION

The analysis of a toy experiment involves synthetic scores generated from $N = 20$ systems, $T = 20$ tasks, and $K = 20$ instances. Each system's performance is modeled by a Gumbel random variable $G_n$ with a center at $\phi \times n$ and a scale of $\beta = 1$, where $\phi$ is a dispersion parameter between 0 and 1. The scores of each system, $s(n, t, k)$, are independent and identically distributed samples of $G_n$ centered at $\phi \times n$ with a scale of $\beta = 1$. Furthermore, the scores from different systems are sampled independently. Since the difference between $G_{n+1}$ and $G_n$ follows a logistic distribution with a mean of $\phi$ and a scale of 1, the probability that system $n + 1$ performs better than system n is at least 0.5, i.e., $P(G_{n+1} - G_n > 0) \geq 0.5$. Thus, the ranking of systems for all k and t is a realization of the true ranking $[1, \cdots, N]$, with a noise term controlled by the dispersion parameter $\phi$. The extreme scenarios are $\phi = 0$ and $\phi = 1$, where $\phi = 0$ means that all scores $s(n, t, k)$ have the same distribution and $\phi = 1$ results in a strong consensus and a clear system ranking. Unless specifically mentioned, each experiment is repeated 100 times for every data point.

### 4.2 ROBUSTNESS TO SCALING

In order to conduct a more detailed comparison of the ranking, we introduce a corruption in the scores of a specific task by rescaling them with a positive factor of $\lambda$. For this experiment, the corrupted tasks are randomly chosen. Although this corruption does not have any impact on our ranking process (since the ranking induced by a task-instance pair remains unchanged), it progressively disrupts the mean aggregation procedure as the value of $\lambda$ increases (see Fig. 3 for detailed results). *This experiment further validates the use of rankings in NLP benchmarking, as these metrics involve different natures of measurements (e.g., BLEU score vs. number of parameters or speed) and can have bounded or unbounded scales.*

### 4.3 PAIRWISE CONFIDENCE ANALYSIS

To determine the number of system comparisons required to achieve a desired confidence level of $\delta$, we use Eq. 4. Fig. 4 presents the results for two confidence levels ($\delta$). The graph illustrates the number of system pairs for which 0.5 is not within the confidence interval, plotted against the number of comparisons for different values of $m$ and $\phi$. As expected, when the rankings are more concentrated (*i.e.*, when $\phi$ is closer to 1), fewer system comparisons are needed to achieve a high number of valid system comparisons. In real-world benchmarks, test sets usually contain more than 500 pairs.

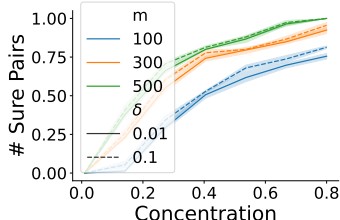

Figure 4: Confidence analysis.

## 5 EMPIRICAL EXPERIMENTS

In this section, we benchmark our methods on real rankings. We introduce a dataset with over 100 million scores, surpassing previous datasets by several orders of magnitude (see Ssec. 5.1 and Ap. C).

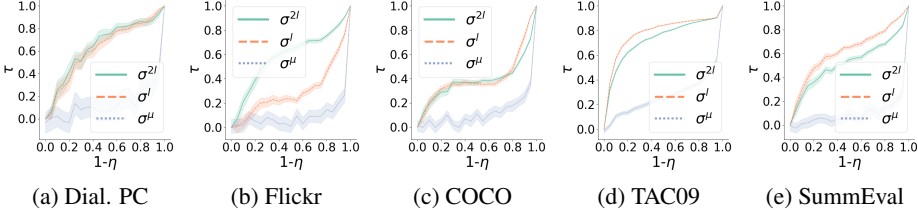

(a) Dial. PC     (b) Flickr     (c) COCO     (d) TAC09     (e) SummEval

Figure 5: Instance-Level Robustness Experiment. We evaluate the robustness of our proposed aggregation methods, namely $\sigma^{2l}, \sigma^l$, and the mean aggregation method $\sigma^\mu$, by randomly removing a proportion $\eta$ of all instances on a specific task for a specific system.

## 5.1 A COMPREHENSIVE COLLECTION OF NLP SYSTEM SCORES

Our dataset builds upon the one used in Colombo et al. (2022b) and includes two types of datasets: those with task-level information and those with instance-level information.

**Datasets with Task Level Information** Our datasets are based on GLUE (Wang et al., 2018), SGLUE (Wang et al., 2019), and XTREME (Hu et al., 2020), which include tasks of varying natures such as accuracy, F1-score, and mean square errors. In addition, we collected data from the GEM Benchmark (Gehrmann et al., 2021), which was an ideal use case for our methods as it encompasses missing data by design (as shown in Table 3 of Gehrmann et al. (2021)) and includes evaluations of various natures such as lexical similarity, semantic equivalence, faithfulness evaluation, diversity, and system characterization (i.e., size of the vocabulary).

**Datasets with Instance Level Information** We did not use the data from Peyrard et al. (2021) for the datasets with instance-level information because they did not provide the sentence and reference test required to add more evaluation metrics or more systems. Therefore, we collected all the data from scratch and extended the dataset in two ways. Firstly, we collected data from five distinct tasks - dialogue (Mehri & Eskenazi, 2020), image description (Young et al., 2014), summary evaluation (Dang et al., 2008; Owczarzak & Dang, 2011; Bhandari et al., 2020; Fabbri et al., 2021), data-to-text (Gardent et al., 2017; Zhou & Lampouras, 2020), and translation (Ranasinghe et al., 2021). For the translation part, we added datasets from WMT15 (Stanojević et al., 2015), WMT16 (Bojar et al., 2016), WMT17 (Bojar et al., 2017), WMT18 (rej Bojar et al., 2018), WMT19 (Barrault et al., 2019), WMT20 (Loïc et al., 2020), and WMT21 (Farhad et al., 2021) in several languages such as en, ru, ts, and others. Secondly, we expanded the set of used metrics from 10 to 17, including Rouge (Lin, 2004), JS (Lin et al., 2006), Bleu (Papineni et al., 2002), Chrfpp (Popović, 2017), BERTScore (Zhang et al., 2019a), MoverScore (Zhao et al., 2019), Baryscore (Colombo et al., 2021b), DepthScore (Staerman et al., 2021), Infolm (Colombo et al., 2021a), CharErrorRate (Morris et al., 2004a), ExtendedEditDistance (Stanchev et al., 2019), MatchErrorRate, TranslationEditRate (Snover et al., 2006), WordErrorRate (Ali & Renals, 2018), WordInfoLost (Morris et al., 2004b), Bleurt (Sellam et al., 2020), and Comet (Rei et al., 2022; 2020). Overall, our benchmark grew *from 250K scores to over 131 M score. This extensive data work is one of the core contributions of this paper, and we believe it will be valuable for future research.*

## 5.2 TASK-LEVEL BENCHMARKING IN REAL-WORLD SCENARIOS

In this section, we explore aggregating missing data with task-level information. First, we test the robustness of our proposed method ($\sigma^l$) against the mean aggregation method ($\sigma^\mu$) and then we quantify the difference between the two output rankings. $\sigma^l$ **is more robust than** $\sigma^\mu$. To compare the effectiveness of aggregation methods in handling missing values on real data, we randomly remove a proportion $\eta$ of the task-level data and measure robustness by computing the Kendall $\tau$ between the rankings of the systems obtained by considering the scores with and without missing values. From Fig. 2, we observe two extreme cases: when no systems are removed (*i.e.*, $\eta = 0$), the aggregation methods output the same value as the one obtained with the full ranking and $\tau = 1$. At the other extreme, when all missing values are removed (*i.e.*, $\eta = 1$), a total absence of correlation can be observed. Overall, we find that $\sigma^l$ achieves a higher correlation, with a large improvement of more than 10 points compared to other methods These results demonstrate that, on average, the rankings remain more stable when using our proposed method.

$\sigma^l$ **outputs a different ranking than** $\sigma^\mu$. We evaluated the correlation between different rankings obtained in the robustness experiment depicted in Fig. 2. Specifically, we compared the rankings

produced by $\sigma^l$ and $\sigma^\mu$ by computing the averaged $\tau$ between the produced rankings when varying the proportion of missing ranking. Results in Tab. 1 show a weak correlation between the two rankings, indicating that they produce different rankings. This weak correlation is further supported by the

results presented in Tab. 2, which measures the percentage of times that the top 1 and top 3 rankings differ when considering the 2k rankings generated in the robustness experiment. *These results demonstrate that in addition to being more robust, our ranking procedure produces different conclusions when benchmarking systems in the presence of missing tasks.*

| | $\tau_{\sigma^l \leftrightarrow \sigma^\mu}$ |
|---|---|
| GLUE | 0.17 ±0.24 |
| SGLUE | 0.33 ±0.27 |
| XTREM | 0.26 ±0.26 |
| GEM | 0.36 ±0.36 |

Table 1: Agreement measured by Kendall $\tau$ correlation.

| Dataset | top 1 | top 3 |
|---|---|---|
| GEM | 0.52 | 0.25 |
| SGLUE | 0.20 | 0.15 |
| GLUE | 0.10 | 0.07 |
| XTREM | 0.19 | 0.09 |

Table 2: Percentage of times the top 1 and top 3 systems are the same between $\sigma^l$ and $\sigma^\mu$.

## 5.3 INSTANCE-LEVEL BENCHMARKING IN REAL-WORLD SCENARIOS

In this section, we evaluate the robustness of $\sigma^{2l}$, $\sigma^l$, and the baseline $\sigma^\mu$.

$\sigma^{2l}$ **and** $\sigma^l$ **are more robust than** $\sigma^\mu$**.** Similarly to the previous robustness experiment, we randomly remove a proportion $\eta$ of scores by discarding all instances of a specific task. *The goal of this missing value sampling is to simulate how missing scores may occur when certain systems are not evaluated on specific tasks.* For each method, Fig. 5 reports the $\tau$ correlation coefficient between the ranking obtained with missing values and the ranking obtained with complete scores.

**Both** $\sigma^{2l}$ **and** $\sigma^l$ **produce highly correlated rankings, while being different from** $\sigma^\mu$**.** We conducted a replication of the agreement analysis presented in Ssec. 5.2 and present the findings in Tab. 3 and Tab. 4. Our results align

| | Corr. |
|---|---|
| $\tau_{\sigma^{2l} \leftrightarrow \sigma^l}$ | 0.80 ±0.22 |
| $\tau_{\sigma^l \leftrightarrow \sigma^\mu}$ | 0.20 ±0.28 |
| $\tau_{\sigma^\mu \leftrightarrow \sigma^{2l}}$ | 0.19 ±0.28 |

Table 3: Agreement.

| | Top 1 | Top 3 |
|---|---|---|
| $\sigma^{2l}$ vs $\sigma^l$ | 0.67 | 0.36 |
| $\sigma^l$ vs $\sigma^\mu$ | 0.21 | 0.09 |
| $\sigma^\mu$ vs $\sigma^{2l}$ | 0.19 | 0.09 |

Table 4: Top 1 and 3 analysis.

with those of our previous experiments, demonstrating that both of our ranking-based procedures ($\sigma^{2l}$ and $\sigma^l$) are more robust in the presence of missing data and yield different rankings than $\sigma^\mu$.

## 5.4 STATISTICAL ANALYSIS

**Confidence interval for practitioners.** The confidence interval is valuable for informing additional comparisons between systems $i$ and $j$. A narrow interval indicates a reliable comparison, while a wider interval suggests more uncertainty and the need for additional comparisons across tasks. For example, in Fig. 6, we report the results of applying $\sigma^l$ on WMT en-de with a confidence level of $\delta = 0.1$. Green value in position $i < j$ illustrate that system $0.5 \notin [\widehat{M}_{ij}^\pi - c_{ij}, \widehat{M}_{ij}^\pi + c_{ij}]$ and $i \succ j$ with high probability. The scale of green displays the distance between 0.5 and the CI, so the greener the more $i \succ j$. The results reveal distinct blocks where top systems (*i.e.*, 9,1,16,15) significantly outperform others with high confidence. Near the diagonal, the elements indicate relatively closer performance of the systems. These findings demonstrate that the confidence interval analysis provides insights into the relative performance of systems.

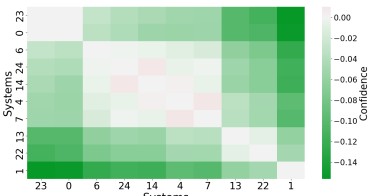

Figure 6: Confidence interval analysis on WMT en-de for a corruption level of $\eta = 0.2$ and a confidence level $\delta = 0.01$. The final ranking can be seen on the x-axis: left to right is best to worst

## 6 CONCLUSIONS AND FUTURE RESEARCH DIRECTIONS

Our study sheds light on the limitations of the conventional mean-aggregation approach, particularly when dealing with missing data. To address this issue, we propose a novel statistical perspective and aggregation procedures that are both robust and grounded in social choice theory. We introduce two alternative methods: the one-level aggregation method ($\sigma^l$) stands out as the most robust approach.

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

# Appendices

# A    EXTENDED RELATED WORK AND OTHER BASELINES METHODS

## A.1    MORE ON MISSING DATA IN NLP.

Another approach to handling missing data in benchmarks would be to create new datasets, however, it can be a sluggish, costly, and expertise-demanding process (see footnote 5 in Lin et al. (2021)). Moreover, there are situations where collection becomes infeasible, such as when working with private datasets, calling for the need to develop tools that can rank systems with missing scores.

## A.2    WHY NOT DIRECTLY IMPUTING DATA IN THE SCORE?

Directly imputing values as scores is not the current practice in NLP. In fact, this approach would be inadequate due to potential variations in metric scale and difficulty, leading to a failure in accurately capturing task difficulty (as mentioned above and in Dolan & Brockett (2005)). To illustrate this to we present some experiments.

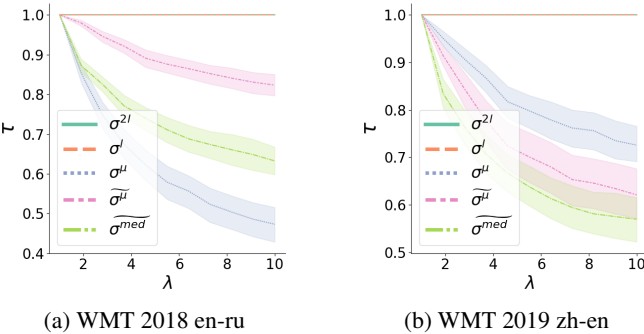

(a) WMT 2018 en-ru          (b) WMT 2019 zh-en

Figure 7: **Imputation methods are not robust to scaling**. To further compare the ranking, we corrupt the scores of a given task by re-scaling them by a factor $\lambda$. Whereas it does not affect our ranking procedure (every ranking induced by a task-instance pair remains the same), it increasingly perturbs the mean aggregation and other imputation procedures as $\lambda$ increases. $\sigma^{\widetilde{med}}$ corresponds to the median imputation and $\tilde{\sigma}^{\mu}$ corresponds to the mean imputation.

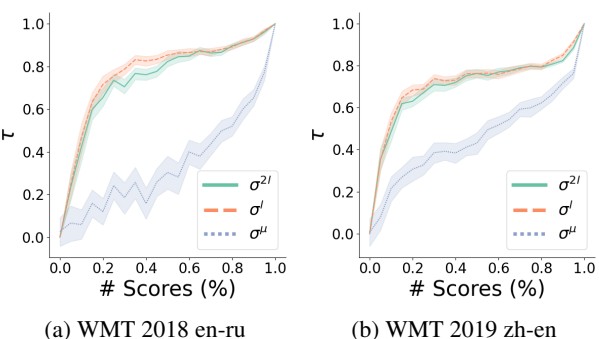

(a) WMT 2018 en-ru          (b) WMT 2019 zh-en

Figure 8: **Additional Instance-Level Robustness Experiment (see Figure 13)**. We evaluate the robustness of our proposed aggregation methods, namely $\sigma^{2l}, \sigma^{l}$, and the mean aggregation method $\sigma^{\mu}$, by randomly removing a proportion $\eta$ of all instances on a specific task for a specific system.

## A.3    ==OTHER IDEAS OF FUTURE WORK==

==In the future, we would like to explore several refinements of the method:==

- **Impact of the inter-task correlation**. The task correlation can impact the choice of the best system. In the future, we would like to study the impact of the choice of the ranking procedure in depth.

- **Impact of misleading evaluation**. Evaluation in NLP can be noisy due to the variety in language and lack of metric robustness Al Sharou et al. (2021); Rodríguez-Cantelar et al. (2023). Future work will include the consideration of this factor when choosing the aggregation method.

- **Comparison with the ANOVA method** St et al. (1989). Although this is slightly outside the scope of the paper, we would like to compare our confidence interval with the one obtained with the ANOVA method.

### A.4 More on the technical contribution of the algorithm.

Our technical contribution boils down to extending the Borda aggregation to the case of missing data (aka incomplete rankings). In the ranking literature, two types of approaches can be identified to deal with partial rankings: Relying on top-k rankings. In this case, all the systems are evaluated but only those that are ranked in the first k positions are provided. There are many methods to aggregate and all in this setting, for example, Ailon (2010). This is different from our scenario where some systems cannot be evaluated on particular tasks. Relying on incomplete rankings. In this case, only k systems are evaluated on a specific task. This fits our scenario. Rank aggregation/statistical analysis in the case of k=2 is called pairwise ranking and is well handled by the literature (Knuth, 1970; Lu & Boutilier, 2014a; Plackett, 1975; Popović, 2017; Zhang et al., 2018). These approaches are limited and only use pairwise comparisons which can lead to paradoxes when ranking more systems. In this paper, we introduce an aggregation procedure for arbitrary values of k. Our main technical contribution is to extend the Borda aggregation to incomplete rankings. To the best of our knowledge, this is the only paper dealing with aggregation -not specifically Borda- of incomplete rankings.

### A.5 Extended Limitations and Perspective for future works

The initial limitation we pinpoint is the task's reliance on the noise model applied to the data, which affects the outcomes. In an extreme scenario where a system lacks all measures, our method might not consistently rank it. Additional edge cases could be investigated, such as a system being poor in only one task with missing data, leading to potentially misleading ranking. To address this, we introduced the confidence interval in Section 3.3, supported by results in Section 5.4, to effectively recognize such challenging scenarios. It's important to highlight that these edge cases can impact all ranking procedures involving missing data.

Another limitation pertains to our ranking procedure's lack of consideration for user preferences regarding tasks. For instance, a user might emphasize certain tasks, such as A, D, and H, with task A carrying greater importance than the others. A natural approach to address this issue involves adopting a weighted variation of the Borda count or drawing inspiration from Dwork et al. (2001). Although this avenue remains unexplored within our current work, it holds promise as a captivating direction for future investigations.

## B Ethical Statement & Limitation of our work

It is important to consider the potential ethical implications and limitations of our work. One ethical concern is the potential bias in the reranking process, as the selection of the "best" hypothesis may favor certain perspectives or reinforce existing biases present in the training data. Care should be taken to ensure fairness and mitigate any potential bias before applying our methods.

## C Dataset Description

### C.1 Task Level Information

We provide additional details on the data collection for Task Level Information.

We gathered data from four benchmark studies, namely GLUE (General Language Understanding Evaluation) (Wang et al., 2018), SGLUE (SuperGLUE) (Wang et al., 2019)[1], XTREME (Hu et al., 2020) and GEM. In the GLUE dataset, there were a total of 105 systems evaluated across nine different tasks: CoLA, SST-2, MRPC, STS-B, QQP, MNLI, QNLI, RTE, and WNLI (Warstadt et al., 2019; Socher et al., 2013; Dolan & Brockett, 2005; Cer et al., 2017; Rajpurkar et al., 2016; Williams et al., 2017b; Dagan et al., 2005; Giampiccolo et al., 2007; Bentivogli et al., 2009; Levesque et al., 2012). The SGLUE dataset consisted of 24 systems evaluated on 10 different tasks: BoolQ, CB, COPA, MultiRC, ReCoRD, RTE, WiC, WSC, AX-b, and AX-g (Clark et al., 2019; De Marneffe et al., 2019; Roemmele et al., 2011; Khashabi et al., 2018; Zhang et al., 2018; Levesque et al., 2012; Pilehvar & Camacho-Collados, 2018). The XTREME benchmark comprised 15 systems and included tasks such as sentence classification (XNLI and PAXS-X), structured prediction (Universal Dependencies v2.5 and Wikiann), sentence retrieval (BUCC and Tatoeba), and question answering (XQuAD, MLQA, TyDiQA-GoldP) (Conneau et al., 2018; Williams et al., 2017a; Yang et al., 2019; Zhang et al., 2019b; Nivre et al., 2018; Rahimi et al., 2019; Pan et al., 2017; Zweigenbaum et al., 2018; 2017; Artetxe & Schwenk, 2019; Artetxe et al., 2019; Rajpurkar et al., 2016; Lewis et al., 2019; Clark et al., 2020).

Each benchmark employed a variety of metrics with different scales, including accuracy, f1, and correlation. Additionally, the GEM benchmark involved 22 systems evaluated using diverse metrics such as prediction length, vocabulary size, entropy, Rouge, NIST, Bleu', Meteor', Bleurt, Nubia, and Bertscore.

## C.2    INSTANCE LEVEL INFORMATION

In this particular setting, our primary focus is on evaluating the performance of natural language generation (NLG) systems, as these scores are among the easiest to collect. We concentrate on five different tasks: summary evaluation, image description, dialogue, and translation. For *summary evaluation*, we utilize the TAC08 (Dang et al., 2008), TAC10, TAC11 (Owczarzak & Dang, 2011), RSUM (Bhandari et al., 2020), and SEVAL (Fabbri et al., 2021) datasets. Regarding *sentence-based image description*, we rely on the FLICKR dataset (Young et al., 2014). For *dialogue*, we make use of the PersonaChat (PC) and TopicalChat (TC) datasets (Mehri & Eskenazi, 2020). For the translation part, we added datasets from WMT15 (Stanojević et al., 2015), WMT16 (Bojar et al., 2016), WMT17 (Bojar et al., 2017), WMT18 (rej Bojar et al., 2018), WMT19 (Barrault et al., 2019), WMT20 (Loïc et al., 2020), and WMT21 (Farhad et al., 2021) in several languages such as en, ru, ts, and others. For all datasets except MLQE, we consider automatic metrics based on S3 (both variant pyr/resp) (Peyrard et al., 2017), ROUGE (Lin, 2004) (including five of its variants (Ng & Abrecht, 2015)), JS [1-2] (Lin et al., 2006), Chrfpp (Popović, 2017), BLEU, BERTScore (Zhang et al., 2019a), and MoverScore (Zhao et al., 2019). For the MLQE dataset, we solely consider several versions of BERTScore, MoverScore, and ContrastScore. Additionally, we incorporate human evaluation, which is specific to each dataset.

## C.3    DATA STATISTICS

To give to the reader a better sense of the richness of our benchmark, we report in Fig. 9 the statistics on our dataset. We demonstrate a diverse distribution of system counts across various datasets, ranging from a minimum of 2 systems to a maximum of 60 systems. Regarding the total number of sentences (instances) and the average number per system, as depicted in Fig. 10 and Fig. 11, the smaller datasets consist of several hundred sentences in total, while the larger datasets encompass up to several hundred thousand sentences in total.

## D    ADDITIONAL REAL-DATA EXPERIMENTS

In this dedicated section, we aim to provide curious readers with a deeper understanding of the capabilities of our methods by presenting additional figures and experimental results. Through these

---

[1]Results can be accessed at https://super.gluebenchmark.com/

supplementary materials, we intend to shed more light on the effectiveness and potential of our approaches, enabling readers to gain valuable insights into our methods.

## D.1 Example of Ranking with missing data on XTREM

In this section, we aim to illustrate the distinction between different rankings obtained using $\sigma^l$ and $\sigma^\mu$ on XTREM dataset for a specific noise realization. Using Tab. 5, we obtain the following rankings:

- $\sigma^l$ gives the following ranking : $M0 > M3 > M2 > M1 > M7 > M5 > M4 > M8 > M9$

- $\sigma^\mu$ gives the following ranking : $M7 > M4 > M0 > M6 > M9 > M2 = M3 > M1 > M8 > M5$.

We can see that the two methods disagree on the best systems in this case. However, as can be seen in our experiments, the ranking-based method is more robust.

| Model | Classification | Structured Prediction | Question Answering | Sentence Retrieval |
|-------|---------------|----------------------|-------------------|-------------------|
| M0 | 90.3 | X | 76.3 | 93.7 |
| M1 | 90.1 | X | 75.0 | X |
| M2 | 89.3 | 75.5 | 75.2 | 92.4 |
| M3 | 89.0 | 76.7 | 73.4 | 93.3 |
| M4 | 88.3 | X | X | X |
| M5 | X | X | X | X |
| M6 | 87.9 | 75.6 | X | 91.9 |
| M7 | X | X | X | 92.6 |
| M8 | X | 75.4 | X | X |
| M9 | 88.2 | 74.6 | X | 89.0 |

Table 5: XTREM dataset with 10 systems and 18 missing values ($\eta = 0.45$)

## D.2 Additional Robustness Experiment on task level datasets

In this section, we report additional experiments on the task level robustness.

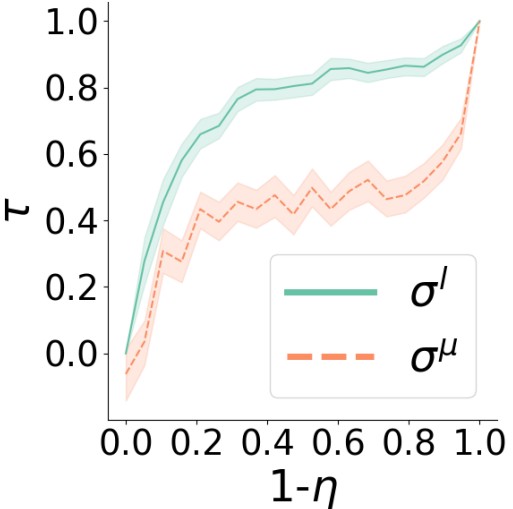

Figure 12: GEM

## D.3 ADDITIONAL ROBUSTNESS EXPERIMENT ON INSTANCE LEVEL DATASETS

In this section, we report additional experiments on the instance level robustness.

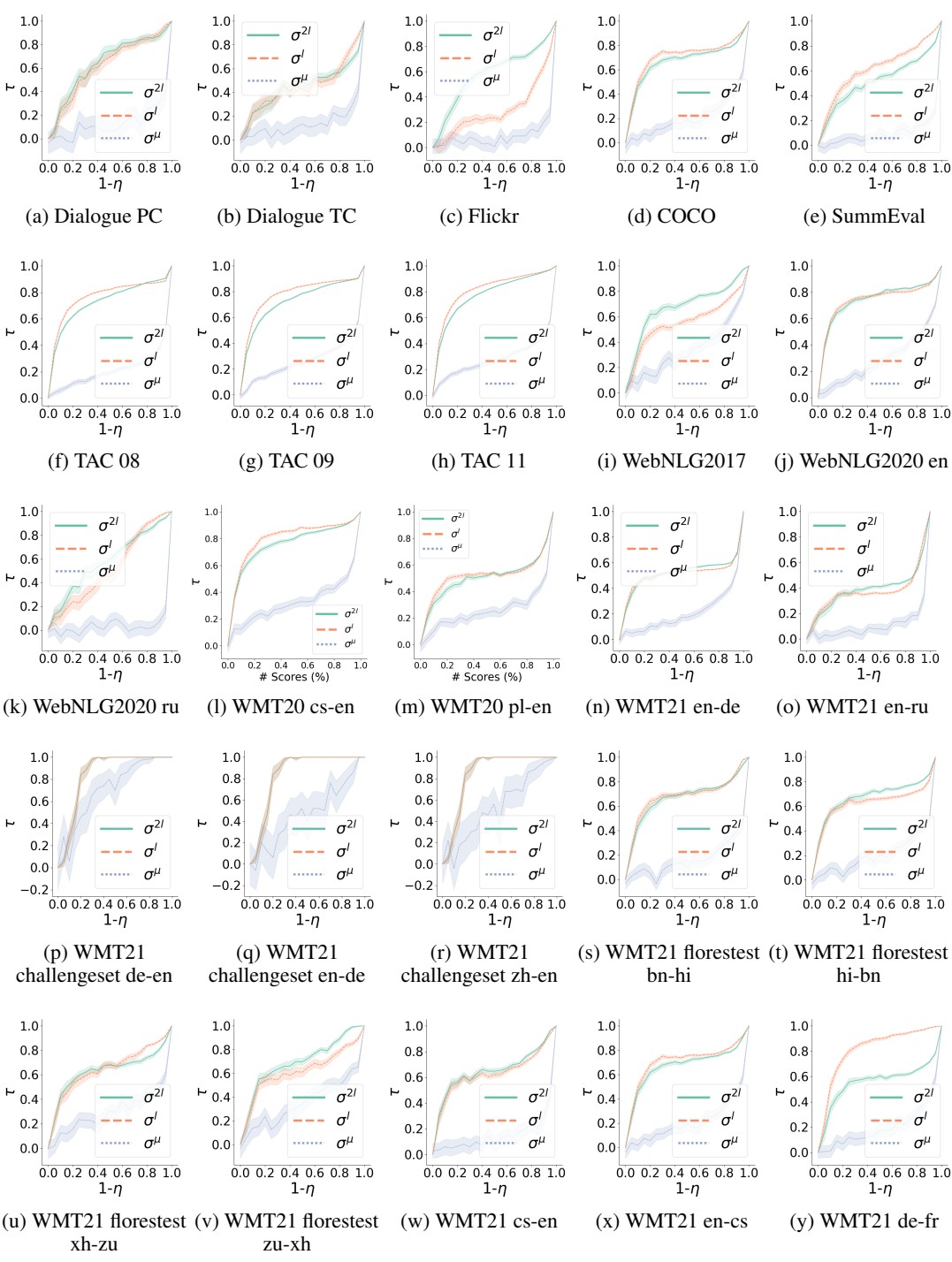

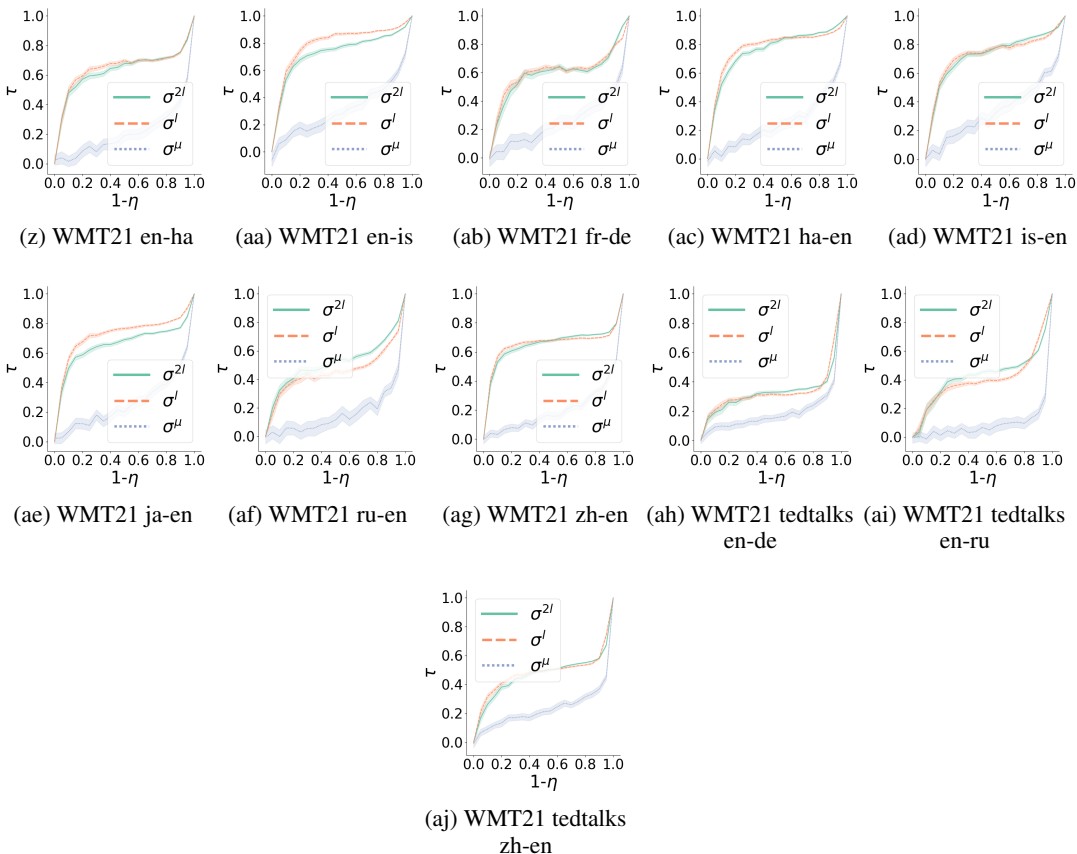

Figure 13: Instance-Level Robustness Experiment. We evaluate the robustness of our proposed aggregation methods, namely $\sigma^{2l}, \sigma^l$, and the mean aggregation method $\sigma^\mu$, by randomly removing a proportion $\eta$ of all instances on a specific task for a specific system. Each experiment is repeated 100 times for each proportion.

## D.4 ADDITIONAL CONFIDENCE ANALYSIS ON TASK LEVEL

In this section, we present additional experiments conducted on four instance-level datasets. We computed confidence intervals for the instance-level, similar to the approach used in Section Ssec. 5.4. Consistent with the main findings in the paper, our observations reveal that closer performance among systems is indicated near the diagonal and we can clearly observe group of systems. This analysis of confidence intervals provides valuable insights into the relative performance of different systems.

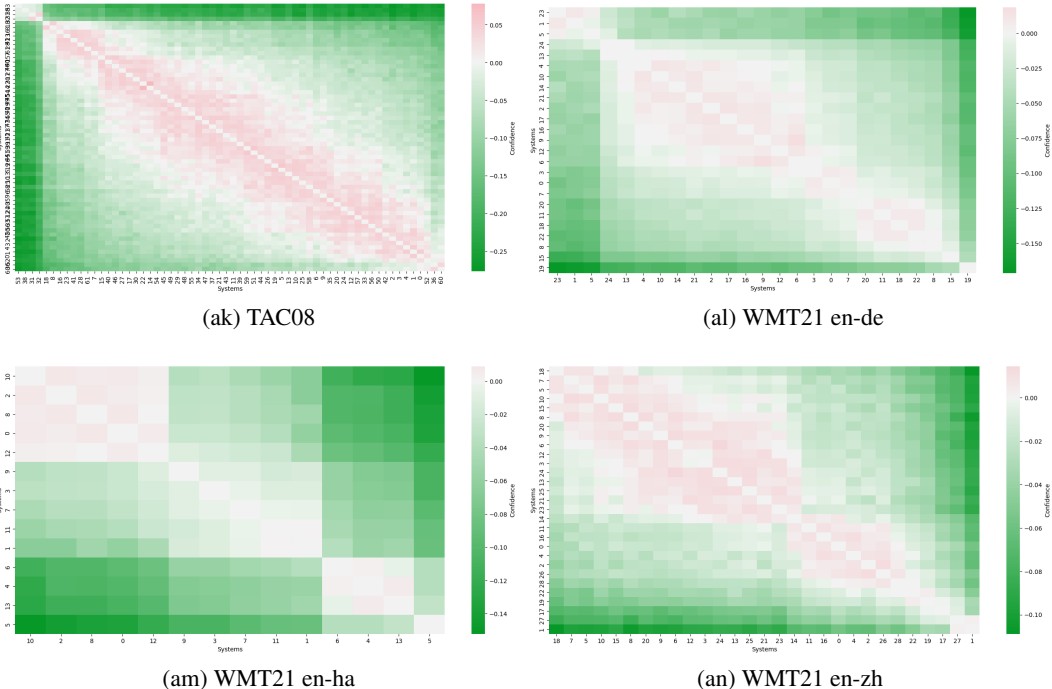

Figure 14: Confidence intervals for various instance level datasets with $\eta = 0.2$ and $\delta = 0.01$

# E  ON THE RANKINGS

This section gathers technical considerations on the ranking methods used in our algorithm.

## E.1  BORDA COUNT ON PERMUTATIONS (IN VECTOR NOTATION)

**Remark 2.** *The Borda count is a ranking system that aggregates a set of permutations $\sigma^1, \ldots, \sigma^L \in \mathfrak{S}_N$ by summing the ranks of each system and then ranking the obtained sums. The procedure is as follows:*

*1. Compute $\mathrm{sum}_n := \sum_{l=1}^{L} \sigma_n^l$ for every $1 \le n \le N$,*

*2. Output $\sigma := \mathrm{Borda}(\sigma^1, \ldots, \sigma^L) \in \mathfrak{S}_N$ that ranks the sums, $\mathrm{sum}_n$ (argsort(argsort(sum$_1$, ..., sum$_T$))).*

## E.2  BORDA COUNT ON PERMUTATIONS IN PAIRWISE MATRIX NOTATION

In Sssec. 3.2.1 we argue that a ranking $\sigma \in \mathfrak{S}_N$ can also be written as a pairwise matrix and in Sssec. 3.2.2 and Sssec. 3.2.3 we further elaborate on how to write ranking data-set $D$ in pairwise matrix form $M^D \in [0,1]^{N \times N}$. Under this notation, the final aggregated ranking $\sigma$ for the Borda count algorithm can be shown to be equivalent to the permutation that sorts the sum of the columns in $M^D$,

$$\sigma = argsort\left(argsort\left[\sum_i M_{i,0}^D, \cdots, \sum_i M_{i,N}^D\right]\right). \tag{5}$$

### E.3 GENERATING ALL COMPATIBLE RANKINGS

In this section, we detail the computation of the $M_{i,j}^\pi$ when item $i$ is not evaluated and item $j$ is evaluated. Let us fix some notation first. For the following, $k$ is the number of observed systems in $\pi$, item $i$ is not evaluated, item $j$ is evaluated and $r$ is the (partial) rank of item $j$. Under this setting, we set $M_{i,j}^\pi = p(n, k, r)$, i.e., the proportion of compatible rankings that rank $i$ before $j$ when $\pi$ has $k$ items. The closed-form expressions for these quantities are given in Eq. 6. Here we note that $t(n, k)$ is the total number of rankings of $n$ items compatible with $\pi$, $S_b^a$ is the number of shuffles of two lists of lengths $a$ and $b$ and $V_b^a$ denotes the variations of $a$ out of $b$ items, i.e., the number of possible arrangements of selections of $a$ objects out of $b$, where the order of the selected objects matters.

$$
\begin{aligned}
p(n, k, r) &= \sum_{i=0}^{n-k-1} V_{n-k-1}^i * (i+1) * S_{i+1}^r (n-k-i-1)! * S_{k-r-1}^{n-k-i-1} / t(n, k) \\
t(n, k) &= (n-k)! * S_{n-k}^k \\
S_b^a &= (a+b)!/(a! + b!) \\
V_b^a &= a!/(b-a)!
\end{aligned}
\tag{6}
$$

**Remark 3.** *A naive algorithm for generating the matrix $M^\pi$ from $\sigma \in S_{N-r_{tk}}$ would have factorial complexity and it is thus exorbitant in practice for a relatively small number of systems, say $N > 10$. However, our solution has a complexity of $O(n^3)$ and can be precomputed once at the beginning of the benchmarking process to efficiently generate the pairwise matrix $M^\pi$ from partial ranking $\pi$.*

### E.4 PROOF OF UNIFORMITY

In this section, we give the intuition and the proof for Eq. 6. This section follows a classic strategy on Enumerative Combinatorics (Stanley, 1986; Wilf, 1999): if we can define an algorithm to generate compatible permutations uniformly at random (such as that in Algorithm 2), we can easily adapt it to count those permutations to yield an efficient counting expression, as we do in Eq. 6.

We start by introducing 2 basic operations of $permute$ and $shuffle$, along with the number of possible outcomes of these operations.

**Permute a list -** $permute(l)$ Given a list of $n$ objects, generate a permutation of these items. There are $n!$ possible ways of permuting $n$ items. An efficient way for generating random permutations is the Fisher-Yates-Knuth algorithm (Knuth, 1970).

**Shuffle two lists -** $shuffle(A, B)$ Given two disjoint lists of distinct elements $A, B$ of lengths $a, b$ respectively, generate a permutation $\sigma$ of the two lists of length $a + b$ in such a way that the relative order of the items in the lists $A$ and $B$ is respected in $\sigma$. This name and idea is based on the popular way of shuffling two decks of cards (Bayer & Diaconis, 1992). Its easy to see that Algorithm 1 generates every possible shuffling with equal probability. The total number of shuffles of lists $A, B$ is given in Eq. 6 as $S_b^a$.

---

**Algorithm 1:** Generate a random shuffle of lists $A$ and $B$

---

1 **for** $i \in [a + b]$ **do**
2     $rand \leftarrow$ random number in $[0, 1]$;
3     **if** $rand > 0.5 \vee B$ *is empty* $\wedge A$ *is non empty* **then**
4         $\sigma(i) = pop(A)$;
5     **else**
6         $\sigma(i) = pop(B)$;
7     **end**
8 **end**

---

**Counting complete, compatible rankings** At this point, we are ready to detail the expression of $p(n, k, r)$ in Eq. 6, both the intuition and the proof of uniformity. For this, we propose in Algorithm 2 to sample complete, compatible rankings and then adapt this sampling algorithm to a counting algorithm in Theorem 1.

**Notation** We start by fixing the notation. Let $\beta$ be a partial ranking of length $k$ which includes item $j$ in rank $r$, $\beta_1 \succ \ldots \succ \beta_r = j \succ \ldots \succ \beta_k$. Let $\eta$ be a disjoint set of $n - k$ items that have not been ranked and which includes the unobserved item $i$. The goal is to generate (i) a compatible ranking with $\beta$ (a ranking $\sigma$ of all the items in such a way that the relative ordering of the items of $\beta$ is maintained) and (ii) which ranks item $i$ before item $j$. We denote the "$s$-head" of a list to the items in the first $s$ positions in that list.

**Intuition** We are now ready to explain the intuition. Each of the possible compatible permutations that rank $i$ before $j$ is generated in the following way:

Algorithm 2 generates permutations that rank item $j$ at position $s$, item $i$ before $j$ and we iterate for all possible values of $s$. First, in line line 2 we select $s - 1$ items randomly from $\eta$, where the order of the items matter (i.e., a variation). Then, we insert item $i$ in a random position of this list, denoted $\eta_{head}$ in line 3. In line 4 we shuffle these two lists, i.e., $\eta_{head}$ and the $r-$head of $\beta$, $\beta_{head}$, i.e., the sublist with the items that are ranked before $j$. The result of the shuffling process is the $s + r$-head of the output permutation $\sigma$. We permute the rest of the unobserved items denoting these list $\eta_{tail}$, in line 6. Finally, we shuffle this list $\eta_{tail}$ and the $k - r$-tail of $\eta$ in line 7. The result of this shuffle is the tail of $\sigma$. Finally, in line 8 we return the concatenation of $\sigma_{head}, j, \sigma_{tail}$, which is clearly a compatible permutation with $\beta$ as the relative order of the items in $\beta$ is maintained in the output.

---

**Algorithm 2:** Generate a random ranking among those compatible with $\beta$

---

1 **for** $s \in [n]$ **do**
2      $\eta_{head} \leftarrow s - 1$ items from $\eta$ where the order matters ;
3      $\eta_{head} \leftarrow$ insert $i$ in $\eta_{head}$ ;
4      $\sigma_{head} \leftarrow$ shuffle$(\eta_{head}, \beta_{head})$ ;
5      $\eta_{tail} \leftarrow \eta \setminus \eta_{head}$ ;
6      $\eta_{tail} \leftarrow$ permute$(\eta_{tail})$ ;
7      $\sigma_{tail} \leftarrow$ shuffle$(\eta_{tail}, \beta_{tail})$ ;
8      **return** $(\sigma_{head} \succ j \succ \sigma_{tail})$ ;
9 **end**

---

It is easy to see that Algorithm 2 generates the target permutations uniformly at random. Following a classic strategy on Enumerative Combinatorics (Stanley, 1986; Wilf, 1999) we use this algorithm as a proof for $p(n, k, r)$.

**Theorem 1.** *The number of complete permutations of $n$ items compatible with partial ranking $\beta$ that rank the unobserved item $i$ before the observed item $j$ is given by the following expression,*

$$p(n, k, r) = \sum_{i=0}^{n-k-1} V_{n-k-1}^i * (i + 1) * S_{i+1}^r (n - k - i - 1)! * S_{k-r-1}^{n-k-i-1} / t(n, k).$$

*Proof.* It is easy to see that in Algorithm 2 there is a bijection between the permutations in the target (that is, the permutations compatible with $\beta$ for which $i \succ j$) and each outcome of Algorithm 2. Clearly, for uniform at random outcomes of the $shuffle$ and $permute$ operations, the outcome of Algorithm 2 will be random as well. Therefore, the number of possible outcomes of the algorithm equals the number of permutations in the target.

It follows that each term in $p(n, k, r)$ Each term in the previous expression comes from a different line in 2:

- Line 2: The number of variations of $i$ items out of $n - k - 1$ is $V_{n-k-1}^i$.

- Line 3: There are $s + 1$ ways of inserting item $i$, thus the term $(r + 1)$.

- Line 4: There are $S_{s+1}^r$ ways of shuffling $\eta_{head}$ and $\beta_{head}$.

- Line 6: There are $(n - k - s - 1)!$ possible permutations of the items in $\eta_{tail}$.

- Line 7: There are $S_{k-r-1}^{n-k-s-1}$ ways of shuffling the two tails.

- Line 8: Finally, since we compute the proportion by dividing among the total number of compatible permutations.

By repeating this process for all $s < n - k - 1$ the proof is completed.

$\square$

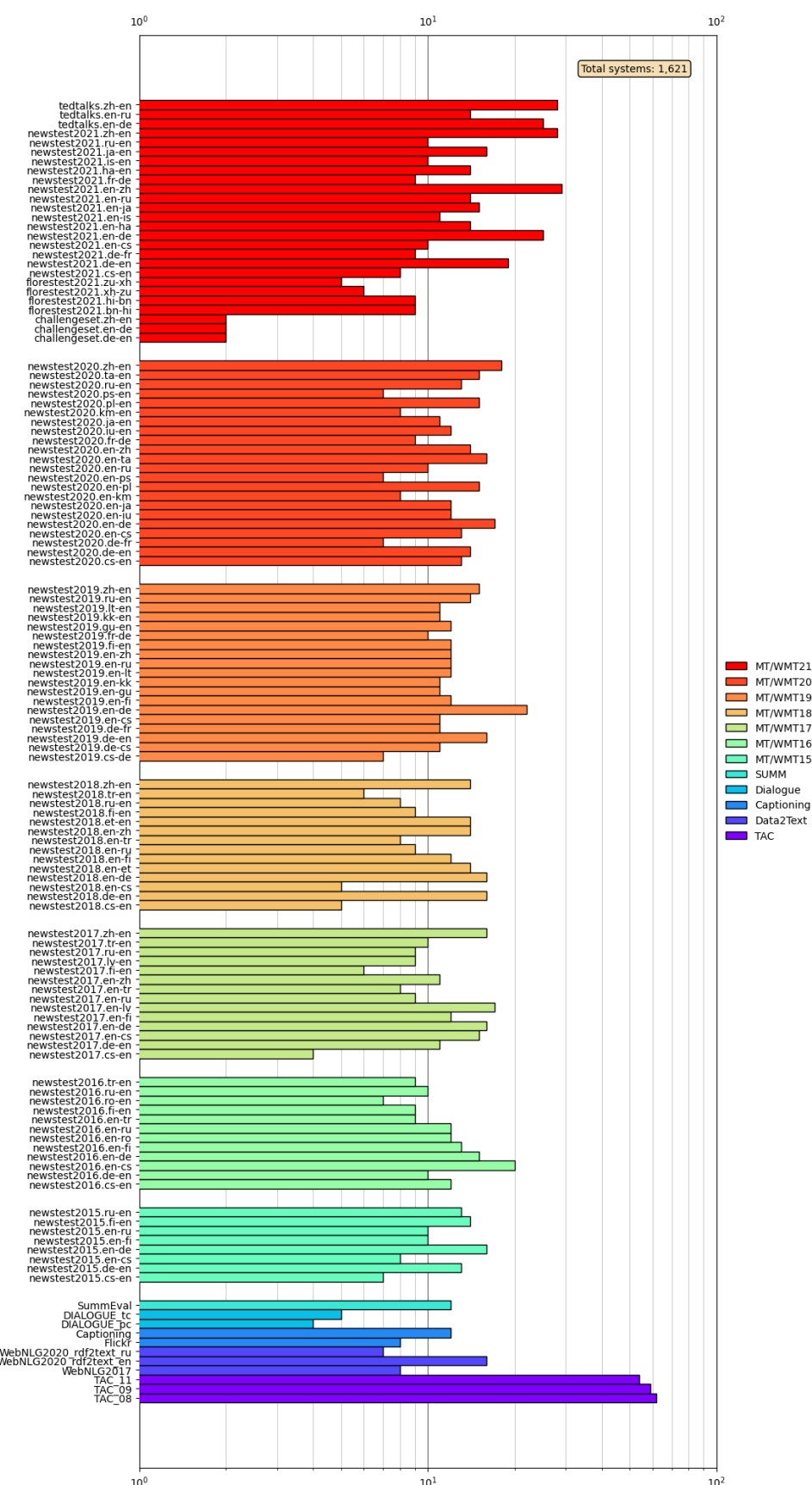

Figure 9: Number of systems in each dataset (log scale)

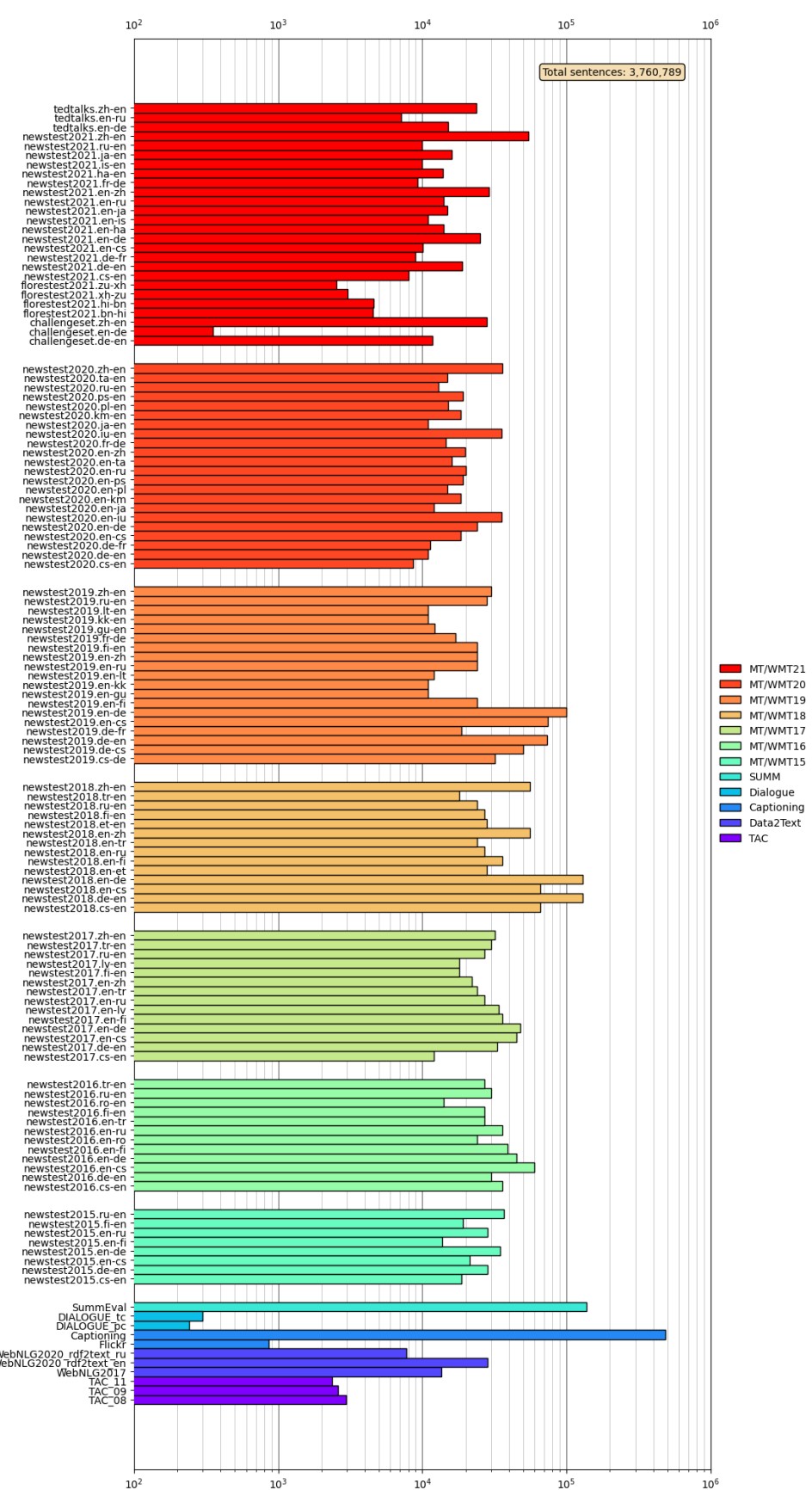

Figure 10: Number of sentences in each dataset (log scale)

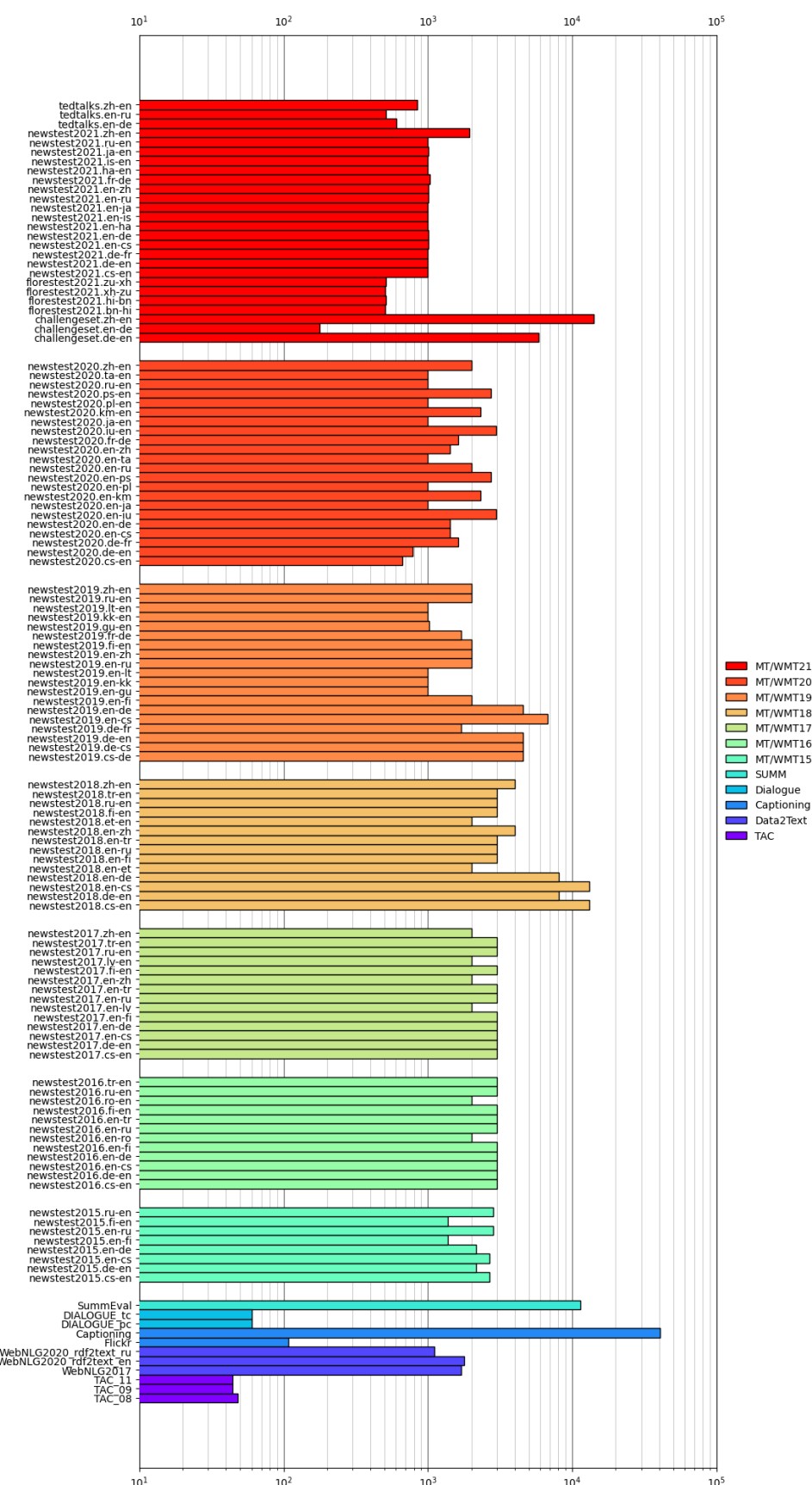

Figure 11: Average number of sentences per system in each dataset (log scale)

