# OpenReview forum: "Towards More Robust NLP System Evaluation: Handling Missing Scores in Benchmarks"
_ICLR.cc/2024/Conference — Submitted to ICLR 2024_

### Official Review · Reviewer_R5ne · 2023-10-29

**Soundness:** 3 good
**Presentation:** 2 fair
**Contribution:** 3 good
**Rating:** 8
**Confidence:** 3

**Summary:**

The paper addresses the problem of combining multiple partial system rankings (from multiple evaluation tasks/instances) to form a single complete ranking. The proposed method consists in building fractional ranking matrices where missing evaluation are replaced by the proportion of permutations compatible with the partial information, and then combining those matrices with the consensus-oriented Borda aggregation method (sum of ranks). The method is adapted to both instance- and task-level aggregations and a O(n³) algorithm is proposed for counting the number of system orderings compatible with a partial rank. Synthetic results on a set of 20 systems, 20 tasks, 20 instances show the potential of the method against a baseline that averages metrics, ignoring missing values. Then, a large set of instance-level and task-level scores is produced and made available for popular benchmarks. Evaluation on this set confirms synthetic data results, however, as noted by the authors, the final rankings produced by Borda aggregation are very different from the mean ranking.

**Strengths:**

The problem of handling large benchmarks with missing evaluations is important because of the cost of running larger and larger benchmarks, and because of the unavailability of evaluation results when systems are closed.

The proposed approach relies on Borda consensus which yields a different outcome to benchmark aggregation.

A O(n³) algorithm for counting the number of permutations that agree with a partial ranking allows completing the ranking matrices.

A large dataset of instance-level evaluation results is released for fostering research in this area.

Both synthetic and real data experimental results are convincing.

The paper is clearly written and easy to read.

**Weaknesses:**

As noted in the paper, Borda aggregation yields very different results from mean aggregation, even in the absence of missing values. This should be investigated before accepting that the resulting rankings are truthful.

The degradation from missing values is the same for Borda and mean aggregations in realistic scenarios with less than 20% missing values, showing a potential lack of interest by practitioners.

Experiment results should be analyzed more thoroughly.

The addressed problem is not NLP-specific although experimental results are restricted to the field of NLP.

**Questions:**

"enables us to leverage the Borda aggregation inheriting its theoretical and practical advantage" => what are they?

Why is sum of ranking matrices (and more generally Borda count) a good aggregation criterion?

How does the method handle misleading evaluation results, when the evaluation metric failed because of the dataset sample bias, or because it is itself an approximation of human evaluation?

Scaling corruption (Fig 3) is not detailed enough. How are the tasks selected for being scaled? How many tasks are being scaled? What is the evaluation metric? Why are there two regimes, starting at 1 or 0 when eta is 0? Font in Figure 3 is also too small

Why is the correlation of sigma_l very different from sigma_2l for some datasets of Fig. 5 while it is very similar for others?
It is not clear from the figures whether # scores (%) is the percentage of removed or kept scores. The discussion is misleading in that regard.

What is the proportion of missing scores in the comparison of rankings by sigma_l and sigma_mu in table 1 and 2?

Confidence analysis (Fig 6) should be compared to other methods such as ANOVA. This section is not very useful to main point of the paper and can be removed in favor of more analysis of previous experiments.

**Details Of Ethics Concerns:**

No concern

---

> ### Author Response · Authors · 2023-11-22
>
> We would like to warmly thank reviewer **R5ne for their review we are glad that they are excited by the paper and find our contribution strong and novel.**
>
> Below we answer the questions of the reviewer:
>
> **1. On the choice of Borda aggregation/lack of ground-truth/theoretical advantages of Borda.** To comment on the validity of using the ranking as ground truth. It's important to recognize the absence of a definitive ground truth ranking. Indeed, the complexity of social choice theory, exemplified by Arrow's impossibility theorem and the Gibbard–Satterthwaite theorem, underscores the inherent challenges in establishing a universally satisfying and consistent ranking system within the specified criteria.
>
> In this paper, we choose to sum the ranking because it is exactly the Borda Count. The Borda Count is a 2 approximation of the Kemmeny consensus [1] and is an excellent choice in practical applications (see [2,3]).
>
>
> 2. **We did not investigate the impact of misleading evaluation results in this paper**, but we do agree this is an interesting question. We did add it to future work (see Appendix).
>
> 3. **On the robustness analysis in the experiments.** $\eta$ varies from 0 to 0.3 and the corrupted tasks are randomly chosen. The change in the final ranking is measured by the Kendall $\tau$. We did add a clarification in the main paper (see updated manuscript).
>
> 4. **In Figure 5.** We agree that the label of the x-axis was misleading. We have updated it by setting it to 1- \eta to make it coherent with the discussion. We attribute the decline in performance on Flickr to the data distribution. However, our investigation into the score distribution did not reveal any discernible differences.
>
> 5. **In Tables 1 and 2**, we quantified the variation in rankings by calculating the average Kendall's tau (τ) between the generated rankings as we varied the proportion of missing rankings. We have included a clarification on this point in the revised version of the manuscript.
> 6. **We added the comparison with ANOVA** in the future work section. We agree with the reviewer this is outside the scope of the paper.
>
> **References**
>
> [1] John G Kemeny. Mathematics without numbers. Daedalus, 88(4):577–591, 1959.
>
> [2] Alnur Ali and Marina Meila. Experiments with kemeny ranking: What works when? Mathematical Social Sciences, 64(1):28–40, 2012.
>
> [3] John J Bartholdi, Craig A Tovey, and Michael A Trick. The computational difficulty of manipulating an election. Social Choice and Welfare, 6(3):227–241, 1989.

---

### Official Review · Reviewer_yPnM · 2023-10-31

**Soundness:** 2 fair
**Presentation:** 2 fair
**Contribution:** 3 good
**Rating:** 5
**Confidence:** 3

**Summary:**

This paper extends a method to rank systems proposed by Colombo et al. (2022) to an incomplete set of scores in tasks (and task instances). The evaluation method is empirically compared to a very simple baseline, with good results. The experiments are performed on a synthetic dataset and an extension of an existing dataset.

**Strengths:**

The main technical contribution of the paper is to extend Colombo et al. (2022) in order to cover for missing task (or instance) scores, via a combinatorial method.

The results are positive in favor of the proposed technique, although the more complex two-level method is not better than the simpler one-level method.

**Weaknesses:**

Originality is low and the contributions weak, as the main contributions are an efficient implementation for a combinatorial problem that allows to extend two pre-existing methods (Colombo et al. 2022) to missing scores, and enlarging an already existing dataset. Unsurprisingly the methods proposed in (Colombo et al. 2022) also are effective in this setting.

The main empirical weakness is that it does not compare to any strong baseline. For instance the baseline that ignores data  using mean aggregation, has too intermingled issues: that of ignoring data and that of using scores from different scales. Thus, from figure 2 it's not clear whether its worse results are caused by one or the other, or, in other words, whether the proposed method is better because it uses ranks (instead of scores) or because it models missing scores. Colombo et al. 2022 already showed that these two methods are better than such a baseline.

The figures have very small fonts, unreadable without extensive zooming.

Minor issues:

* Fig 3 is not readable, same with other figures
* Reference missing in: "We did not use the data from [4]"
* Citations use the wrong format "authors (year)" instead of "(authors, year)"

**Questions:**

In section 4, it seems that the toy experiment is only applied to robustness scaling and pairwise confidence analysis, but as I started to read I was expecting more development experiments. Could you mention why you only check those two factors and not other?

From figure 5, it would seem that instance level information is not helpful and is harmful in three datasets, but there is no explicit elaboration on this (only a brief mention in the conclusions).

---

> ### Author Response · Authors · 2023-11-22
>
> We would like to warmly thank reviewer qUQT for their review. The main concerns of the reviewer are focused on the novelty and the choice of the baseline.
>
> **On the novelty.**
>
> The first novelty of the paper is to identify and formalize the problem of benchmarking in the presence of missing data. Previous work in NLP often simply ignores the missing data when choosing the best system (see Pfeiffer et al., 2022; Lin et al., 2022; Martin et al., 2020; Guibon et al., 2021; Peng et al., 2019). This is mainly due to the difficulty of collecting data/private datasets or more recently the cost of running expensive models such as GPT.
>
> To the best of our knowledge, this is the first paper to tackle the problem of benchmarking with missing data in the NLP community.
>
> **On the technical contribution.**
>
> Our work builds on Colombo et al 2023. However, the addressed problem is different as such their method cannot be naively applied. Second, it seems that the reviewer missed the technical aspect of our paper. Specifically:
> we introduce a new ranking algorithm for missing data. Our newly designed method can be adapted to both instance- and task-level aggregations. This method is not trivial and relies on counting the number of system orderings compatible with a partial rank. We respectfully refer the reviewer to Section 3. Our algorithm’s complexity is polynomial $O(n^3)$ while the naive version would be factorial. Additionally (see response to cmy7) our estimator is unbiased.
> We introduce confidence intervals for the ranking see Section 3.3 to better understand the uncertainty linked to the ranking.
>
> Additionally, we conducted an extensive data collection and gathered a new dataset with a public release. Our effort is crucial for future work. Concretely, our dataset gathered over 131M score which is an order of magnitude larger than all existing datasets.  We believe that this will spur more research in benchmarking, a critical area in NLP, especially considering the surge in Generative Models.
>
> **On the choice of the baseline:**
> To the best of our knowledge, this is the first work that addresses the issue of benchmarking in the presence of missing data for NLP systems. This is indeed a strong contribution to the paper.
>
> **On the readability issue.**
> We have diligently incorporated the reviewer's feedback into our manuscript. This includes rectifying typos, addressing citation issues, and increasing the size of figures as suggested.
>
>
> _We answer the questions of the reviewer below:_
> 1. **On the toy experiments.** The selection of these two factors, namely robustness to scaling and confidence, stems from the specific demands of NLP evaluation. In the realm of generative models, NLP practitioners frequently encounter metrics on diverse scales, some of which may even be unbounded, as exemplified by BartScore. Furthermore, practitioners often neglect confidence intervals, making this paper's focus on them a notable contribution. Given the already dense nature of the paper with numerous experimental results, our decision to explore additional factors, such as robustness to noise in real data experiments, serves to further substantiate the efficacy of our approach on authentic rankings.
>
> 2. **On the difference between $\sigma^l$ and $\sigma^{2l}$.** It's important to recognize the absence of a definitive ground truth ranking. Indeed, the complexity of social choice theory, exemplified by Arrow's impossibility theorem and the Gibbard–Satterthwaite theorem, underscores the inherent challenges in establishing a universally satisfying and consistent ranking system within the specified criteria.
> However, it is worth noting that the proposed method is more robust than the widely adopted method (namely the mean aggregation in NLP) on all the considered datasets.
>
> We hope our answers address all concerns of reviewer yPnM and **we hope they would be keen to consider raising their score.**

---

> > ### Comment · Area_Chair_MgBi · 2023-12-03
> >
> > Hi reviewer yPnM. Can you kindly check the authors' rebuttal to the weaknesses you pointed out?

---

### Official Review · Reviewer_qUQT · 2023-11-01

**Soundness:** 2 fair
**Presentation:** 3 good
**Contribution:** 3 good
**Rating:** 6
**Confidence:** 3

**Summary:**

This paper tackles the problem of system benchmarking with some scores missing. The proposed approach utilizes a compatible partial ranking approach to impute the missing data and use the Borda count method to do the aggregation. Two scenarios are considered, task-level or instance-level scores are available. The evaluation is done by comparing the system ranking against the groundtruth of complete results.

**Strengths:**

* Tackles the important task
* The proposed approach empirically outperforms the baseline
* Both task-level and instance-level evaluations are covered

**Weaknesses:**

* Lack of closer looks at the correlation between tasks, since similar tasks might be "easier" to predict

**Questions:**

* Are there any other stronger baselines or previous works to compare with?

---

> ### Author Response · Authors · 2023-11-22
>
> We would like to warmly thank reviewer qUQT for carefully reading our manuscript and for their enthusiasm about our work. We indeed hope that our work will be widely adopted by the community as we firmly believe it provides a more robust way to evaluate NLP systems.
>
> Below is a response to the reviewer's question:
> 1. **About the baselines.** To the best of our knowledge, this is the first work that addresses this issue of benchmarking in the presence of missing data for NLP systems. This is indeed a strong contribution to the paper.
> 2. **About the correlation between tasks.** Numerous studies in the literature explore the correlation between metrics (see [1] for example). We did consider this aspect in the early stages of the real data analysis, however, delving into it extensively proved challenging. Additionally, given the amount of experiments in the paper, we believe this would require work on its own. We do agree this would be an interesting follow-up work and added it to the paper's next research directions (see updated version of the manuscript). A promising avenue for studying this problem could lie in investigating confidence intervals. _If all metrics exhibit strong correlations, it could lead to a reduction in the size of the confidence interval—a potential starting point for a more in-depth examination._
>
> **References:**
>
> [1] Colombo, P., Peyrard, M., Noiry, N., West, R., & Piantanida, P. (2022). The glass ceiling of automatic evaluation in natural language generation. Findings AACL 2023.
>
> We hope our answers address **all concerns of reviewer qUQT and we hope they would be keen to consider raising their score.**

---

### Official Review · Reviewer_cmy7 · 2023-11-01

**Soundness:** 3 good
**Presentation:** 2 fair
**Contribution:** 3 good
**Rating:** 5
**Confidence:** 4

**Summary:**

This submission addresses the significant and increasingly relevant problem of benchmarking on multiple datasets when not all systems have been run on all tasks, or even all instances in a task. They propose a novel formalism to derive system rankings (with confidence) from results missing some scores, and show that this improves robustness compared to simply averaging over only available scores.

**Strengths:**

This is clearly a relevant problem -- benchmarking general-purpose models is increasingly done by comparing results on multiple datasets and tasks, but due to many reasons (outlined in the paper) not all systems may be run on all tasks, which makes simple averaging impractical.

The method is clever. It combines 1) estimates based on the proportion of total orders with a given pairwise ordering that are compatible with the observed partial ordering, 2) Borda count on the task orderings into a final ranking, 3) confidence intervals on the resulting rankings.

The formalisation of the problem is clear and useful, and a lot of detail is provided in the appendix. One of the contribution is a practical, non combinatorial method solving the non-trivial problem of estimating the proportion of total orders compatible with an observed partial order.

The method seems to yield much improved robustness compared to simple averaging, and the resulting ranking remains much closer to reference ranking when the proportion of missing scores increases.

**Weaknesses:**

Although the methodology is well described overall and there is a lot of useful detail in the paper and the (extensive) appendix, the motivations are sometimes lacking. For example, is averaging still the right way to combine estimated ranks? Also, imputation methods usually don't use naive distribution estimate, but try to leverage observed data to improve the missing data imputation -- e.g. if scores are missing for systems i and j on a given task, but i usually outperforms j whenever they are both observed, it seems sub-optimal to set M_ij to 0.5 (step 2, p. 5).

The paper feels rushed at times and there are lots of readability issues, including with the notation (see below).

This is a substantial paper with a lot of material. The downside is that it is hard to pack that much material in 9 pages, and difficult to follow the paper without the appendices. There seems to be simply too much material re. experimental results in the last three pages. As a consequence, the Figures are mostly unreadable or unclear and the experimental section does not do a good job supporting the arguments and conclusions of the paper.

To be clear, I think this is an interesting paper with significant results, but the presentation does not do it justice.

**Questions:**

It was not fully clear why the 'argsorts' are systematically doubled (p.3, p.6). E.g. in Eq. 3, it seems that computing the average of estimated scores, one sort would be enough to recover the permutation with correct ranking?

Clarity:
* "input" in Sec. 3.2.1 is likely "impute" (the missing data/information)?
* Still Sec. 3.2.2: p_{i,j} pops up in the last paragraph -- is that M_{i,j}?
* Sec 3.2.3, step 3.: Need some reference to a publication or appendix for Borda count aggregation
* Figures are overall way too small and often unreadable. Their positioning is odd, for example Fig. 2 (top p.7) is referenced on p.9.
* The x-axis in Fig 2 and Fig 5 seem to show the proportion of scores observed rather than proportion of scores removed. As described in the text, Kendal Tau tends to 1 when there is no (0%) missing data.
* What is "[4]" in Sec. 5.1?
* Sec 5.2: "in the robustness experiment" -> not clear what you mean by that and where those are described.

Typos:
* Citations lack brackets in most places -- likely an issue with \cite[] usage with the style file
* p.2: "Our includes..."
* p.3: "previously mentioned." ... article?
* p.3: "on a k of test instances" -> on k test instances?
* p.5, l-4: Superscripts of M seem messed up
* p.9: "in Ssec 5.3" is likely 5.2 (we are in 5.3)

---

> ### Author Response · Authors · 2023-11-22
>
> We thank reviewer cmy7 for their careful reading of the manuscript. We are glad they acknowledge that they found the problem interesting and impactful. We are particularly thrilled by their positive assessment, like our technical contribution, **and find that “the method is clever”.**
>
> Below is a response to the reviewer's concerns:
>
> **1. On the method.**
> The reviewer expresses concern regarding the appropriateness of assuming uniformity for unobserved rankings. In response, we demonstrate that in the realm of ranking data, unlike real-valued data, the estimation process remains consistent despite considerable noise.
> The conventional proofs affirming the consistency of the estimator for the Borda algorithm under various assumptions rely on demonstrating that for any pair $i,j$, if the data is drawn from a model satisfying the strong-unimodality property (such as Plackett-Luce or Mallows model just to name a few popular ones) with a median $\sigma_0$ for which  $\sigma_0(i)<\sigma_0(j)$, then $\mathbb{E}[\sigma(i)] < \mathbb{E}[\sigma(j)]$, and thus, the Borda algorithm accurately ranks such pairs [1,2,3]
> Here, we demonstrate that if Borda accurately ranks the pair $i,j$ without noise, i.e., if $\mathbb{E}[\sigma(i)] < \mathbb{E}[\sigma(j)]$, then it will also correctly rank $i$ and $j$ in expectation, given an equal noise rate $\eta$ for both $i$ and $j$. To see this, note that the expected imputed ranking value will be the average of all possible rankings, $0.5n$. Therefore,  the expected ranking of $i$ will be equal to $\mathbb{E}[0.5n\eta + (1-\eta)\sigma(i)] = 0.5n\eta + (1-\eta)\mathbb{E}[\sigma(i)] $, where the equality holds by linearity of the expectation. By noting that $\mathbb{E}[\sigma(i)]\mathbb<{E}[\sigma(j)]$ implies $0.5n\eta + (1-\eta)\mathbb{E}[\sigma(i)] <0.5n\eta + (1-\eta)\mathbb{E}[\sigma(j)] = \mathbb{E}[\sigma(i)] < \mathbb{E}[\sigma(j)]$ we conclude the proof.
>
> **References:**
>
> [1] Irurozki, E., Perez, A., Lobo, J., & Ser, J. del. (2021). Online Ranking with Concept Drifts in Streaming Data. Joint European Conference on Machine Learning and Knowledge Discovery in Databases.
>
> [2] Fligner, M. A., & Verducci, J. S. (1988). Multistage Ranking Models. Journal of the American Statistical Association, 83(403), 892–901. https://doi.org/10.2307/2289322
>
> [3] Caragiannis, I., Procaccia, A. D., & Shah, N. (2013). When Do Noisy Votes Reveal the Truth? Proceedings of the Fourteenth ACM Conference on Electronic Commerce, 143–160. https://doi.org/10.1145/2482540.2482570
>
> 2. **On the readability issue.**
> We've invested considerable effort in meticulously collecting data, resulting in a robust experimental contribution that makes the paper dense. In response to the reviewer's valuable feedback, we've implemented changes, relocating figures to the appendix and meticulously addressing all their suggestions. This includes rectifying notations, refining legends, correcting typos, and ensuring the accuracy of citations. All the modifications are highlighted in yellow in the updated paper.
>
> 3. **On the two argsorts.**
> The argsorts are used to convert the combined scores of the systems into rankings. For instance, scores [1.0, 3.5, 2.0, 2.2] would be transformed into [0, 3, 1, 2], indicating that system 1 is ranked first, system 2 is ranked fourth, system 3 is ranked second, and system 4 is ranked third. The initial argsort provides an ordering of systems based on their scores, while the second one generates rankings of the systems relative to this ordering.
>
> We hope our answers address all concerns of reviewer cmy7’s and we hope they would be keen to consider raising their score.

---

> > ### Comment · Area_Chair_MgBi · 2023-12-03
> >
> > Hi reviewer cmy7. Can you kindly check the authors' rebuttal to the weaknesses you pointed out?

---

### Meta-Review · Area_Chair_MgBi · 2023-12-10

**Metareview:**

This paper addresses the challenge of benchmarking systems on multiple datasets when not all systems have been evaluated on all tasks or instances within a task. The authors propose a novel formalism that combines estimates based on the proportion of total orders compatible with observed partial orderings, Borda count, and confidence intervals to derive system rankings with improved robustness compared to simple averaging.

Strengths: The strengths of the paper lie in addressing a relevant problem, presenting a novel methodology, and providing a clear formalization of the problem. The proposed method demonstrates improved robustness in system rankings, particularly as the proportion of missing scores increases.

Weaknesses:
1. Motivations for certain aspects of the methodology, such as using averaging to combine estimated ranks and the use of naive distribution estimates, are not adequately explained.
2. The paper has some readability issues. The extensive material in the last three pages, especially regarding experimental results, makes figures largely unreadable and weakens the support for the paper's arguments and conclusions.
3. The empirical comparison lacks strong baselines, and concerns are raised about the clarity of the correlation between tasks, potentially impacting the interpretation of results.

**Justification For Why Not Higher Score:**

While the paper addresses an important problem and makes a substantial contribution, the presentation issues and concerns about motivations and methodology should be addressed.

**Justification For Why Not Lower Score:**

n/a

---

### Decision · Program_Chairs · 2024-01-16

Reject